# Optimizing Cycle Life Prediction of Lithium-ion Batteries via a Physics-Informed Model

**Constantin-Daniel Nicolae**[*]                                        *cdn27@cam.ac.uk*
*University of Cambridge*

**Sara Sameer**[*]                                                      *sarasameer991@gmail.com*
*National University of Computer and Emerging Sciences*

**Nathan Sun**[*†]                                                      *nsun@college.harvard.edu*
*Harvard University*

**Karena Yan**[*]                                                       *kyan@princeton.edu*
*Harvard University*

**Reviewed on OpenReview:** *https://openreview.net/forum?id=1weZ9Wsajk*

## Abstract

Accurately measuring the cycle lifetime of commercial lithium-ion batteries is crucial for performance and technology development. We introduce a novel hybrid approach combining a physics-based equation with a self-attention model to predict the cycle lifetimes of commercial lithium iron phosphate graphite cells via early-cycle data. After fitting capacity loss curves to this physics-based equation, we then use a self-attention layer to reconstruct entire battery capacity loss curves. Our model exhibits comparable performances to existing models while predicting more information: the entire capacity loss curve instead of cycle life. This provides more robustness and interpretability: our model does not need to be retrained for a different notion of end-of-life and is backed by physical intuition.

## 1 Introduction

Predicting the cycle life of a lithium-ion battery remains challenging due to the complexity of the chemical side effects responsible for degrading the performance of a battery as it is repeatedly cycled. In particular, it is well known that solid electrolyte interphase (SEI) formation crucially affects cycle life and occurs within the first few charging/discharging cycles (von Kolzenberg et al., 2020; Wang et al., 2011; Spotnitz, 2003; Broussely et al., 2001). Accurately predicting the cycle life of batteries while accounting for all these side chemical processes is important for maintaining battery performance.

Recently, Severson et al. (2019) presented a state-of-the-art dataset containing 124 lithium-ion batteries with 72 different fast-charging policies and showed that a regularized linear regression model predicting cycle lifetimes performs very well on batteries with different charge policies. They also successfully showed that this prediction can be obtained within the first hundred charging cycles. Their method of using early-cycle data for prediction has great practical implications, since one need not wait to charge a battery for many cycles before knowing its lifetime.

Previous literature on predicting cycle lifetimes of batteries is rich. Data-driven models (Yao et al., 2022; Celik et al., 2022; Abu-Seif et al., 2022; Xing et al., 2023), which focus on using machine learning techniques to identify trends in how batteries degrade, have been thoroughly studied, with models ranging from linear models (Severson et al., 2019) to neural networks (Celik et al., 2022; Strange & Dos Reis, 2021; Su et al.,

---

[*]All authors contributed equally.
[†]Corresponding author.

2021) to support vector regression (Zhu et al., 2022). These types of models are agnostic to the mechanisms of degradation, but require significant effort to fine-tune, as their performance depends heavily on hyperparameter optimization, feature selection, and the quality of the training data. Physics-based models, which rely upon knowledge in how a battery degrades over time, have also been well-studied. These methods usually rely upon cell chemistry (Wright et al., 2003; Rahman & Lin, 2022; Yang et al., 2017) or analyzing how the material in the electrodes changes over time (Christensen & Newman, 2004; Pinson & Bazant, 2012). However, since batteries can be charged/discharged in a variety of environments, this hinders how descriptive physical models by themselves can be. Recently, hybrid models combining physics knowledge with a data-driven approach have been proposed (Xu et al., 2022; Cordoba-Arenas et al., 2015; Pang et al., 2022; Saxena et al., 2022) to combine the advantages of both approaches.

We propose a hybrid model combining a physics-based equation and a self-attention mechanism for prediction. The latter is inspired by the recent rise of transformers to predict sequential data (Nguyen et al., 2022; Vaswani et al., 2017), and the former uses physics insights to capture more information on the behavior of the capacity loss curves of lithium-ion batteries. Although transformers have already found applications in predicting the cycle life of batteries (Xu et al., 2023), combining a physical equation with self-attention to predict complete capacity loss curves has not been thoroughly studied to the best of our knowledge.

Using the hybrid model, we are able to predict battery capacity loss curves. Cycle life is defined as the point where the effective capacity of the battery has dropped to a percentage of its nominal capacity. In the literature, 80% is typically the threshold, but other studies have considered 85% (Schauser et al., 2022; Sours et al., 2024). With entire capacity loss curves, we are able to predict cycle life for all other thresholds without retraining, while models that are specifically trained with the 80% threshold must be retrained. This accounts for conditions that may require a different threshold for cycle life, such as the make of the battery and the presence of non-ideal operating conditions.

The paper is organized as follows. In Section 2 we introduce data processing techniques and extract relevant information from discharge curves. We then introduce a hybrid model utilizing an Arrhenius Law to model capacity loss and utilize self-attention to predict cycle life from early-cycle discharge data. Finally, we compare our errors with Severson et al. (2019) and conclude with future directions.

## 2 Data Processing

We utilize the public dataset provided by Severson et al. (2019). This dataset comprises 124 lithium-ion phosphate/graphite battery cells, each with a nominal capacity of 1.1 ampere-hours (Ah) (A12, 2014). The cells are cycled (repeatedly charged/discharged) until end of life, which we take as the point where the effective capacity of the battery has dropped to 80% of its nominal value. The ratio of the effective and nominal capacity is commonly referred to as the state of health. All battery cells in this dataset are cycled under fast-charging conditions in a constant-temperature environment; however, the charging policy dictating the specific charge rate schedule differs from cell to cell.

In our work, we preserve the train/test/secondary test data split in Severson et al. (2019), allowing for a direct comparison between our result and theirs. The primary test set was obtained using the same batch of cells as the train set and similar charge policies; therefore we use it to evaluate the model's ability to interpolate in the input space. On the other hand, the secondary test set was obtained from a different batch of cells and using significantly different scheduling, and we use it to examine the model's ability to extrapolate. Ensuring our model can generalize to different batteries makes for an advantage over prior work (Saxena et al., 2022; Strange & Dos Reis, 2021) that uses a random split of cells into train/validation/test sets.

For each cell, four forms of data are recorded:

1. **Cycle life:** The number of charge/discharge cycles until the state of health drops to 80%, ranging from 150 to 2,300.

2. **Charge policy:** The schedule of charge rates followed during cell cycling.

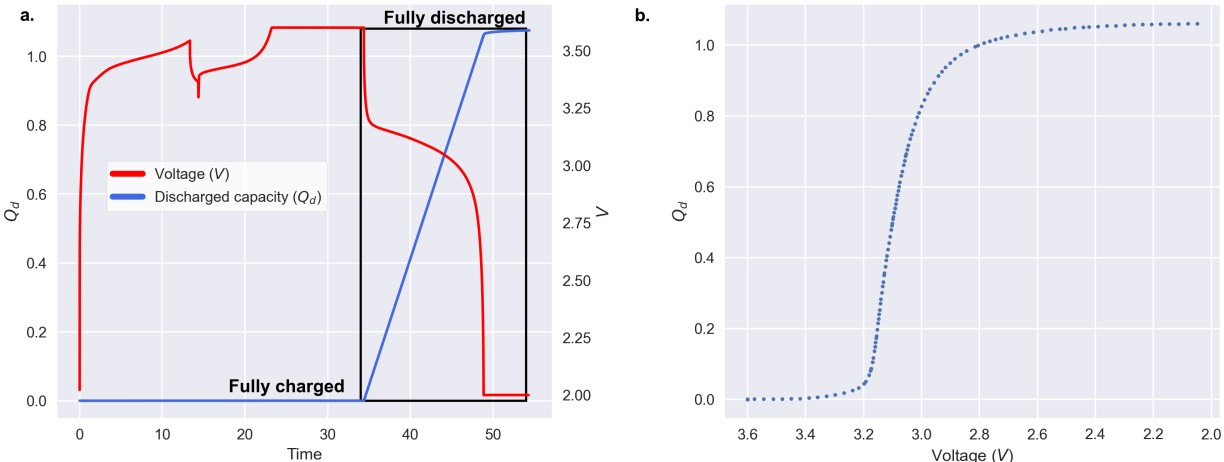

Figure 1: (a) Discharged capacity, $Q_d(t)$, and voltage, $V(t)$, measured for one battery over the course of a charge-discharge cycle. The discharge portion of the cycle is boxed in black. (b) The discharge-voltage curve, $Q_d(V)$, for the same battery. Note that the x-axis in (b) is flipped to reflect voltage decreasing over the course of the discharging process.

3. **Cycle summary features:** Features calculated for each cycle, such as the state of health, internal resistance ($IR$), average cell temperature ($T_{\text{avg}}$), and maximum cell temperature ($T_{\text{max}}$).

4. **Full cycle data:** Measurements taken over the course of each cycle, such as voltage ($V$), discharged capacity ($Q_d$), and temperature ($T$).

Voltage ($V$) and discharged capacity ($Q_d$) are particularly relevant for lifetime prediction and are depicted for an example charge-discharge cycle in Figure 1(a).

The discharge-voltage curve for a cycle, denoted $Q_d(V)$, is constructed by plotting $Q_d$ against $V$ for the discharge portion of the cycle, as boxed in Figure 1(a). One such curve is shown in Figure 1(b). According to manufacturer specifications, a battery cell is considered fully charged when its voltage reaches 3.6V and fully discharged when it reaches 2.0V (A12, 2014). An analogous curve can be constructed for each cycle of a battery's operation.

Choosing the voltage to be our $x$ coordinate for the discharge curves results in a highly irregular set of evaluation points for $Q_d(V)$. As illustrated in Figure 1(b), the $V$ values sampled at regular timesteps are sparser in the intervals [3.2V, 3.6V] and [2.0V, 3.0V] and denser in the interval [3.0V, 3.2V]. Moreover than being irregular, the points at which $Q_d(V)$ were to be evaluated are also different from one cycle to another. This makes direct comparisons between different cycles difficult. To standardize the discharge curves, we utilize radial basis function interpolation, a standard method used for unstructured inputs. In 1D, this consists of an interpolant given by Equation 1:

$$\hat{Q}_d(V) = p_m(V) + \sum_{i=1}^{N} \lambda_i \phi(|V - V_i|) \tag{1}$$

The datapoints are therefore interpolated by a weighted sum of radial basis functions, with the origins at the interpolation nodes $V_i$, and by $p_m(V) = c_0 + c_1 V + \cdots + c_m V^m$, a polynomial of degree $m$. The coefficients of $p_m$ and the weights $\lambda_i$ are found by solving a set of linear equations. The form of $\phi$ dictates the final form of the interpolant. The role of the polynomial term is to model any possible trend in the data, which the RBFs are unable to do due to their symmetric, radial nature.

The evolution of $Q_d(V)$ over cycles can be exploited to predict battery lifetime. Severson et al. (2019) observe that more dramatic early-cycle curve sagging occurs for batteries with low lifetimes than for batteries with

high lifetimes, as visualized in Figure 2(a, b). To capture the phenomenon of curve sagging, Severson et al. (2019) propose taking the discharge-voltage curves for cycles 100 and 10 and computing the difference between the two. This new curve, calculated as $Q_{d,100}(V) - Q_{d,10}(V)$, is denoted $\Delta Q_{100-10}(V)$. Figure 2(c,d) shows that $\Delta Q_{100-10}(V)$ succinctly capture the difference in curve sagging behavior between two batteries of different cycle lives, and Figure 2(e) shows a clear linkage between curve shape and cycle life across all batteries in the dataset. In particular, batteries with shorter cycle lives exhibit more ample dips in the $\Delta Q_{100-10}(V)$ curve.

Now statistical quantities, such as the variance, minimum, and mean, of $\Delta Q_{100-10}(V)$ are calculated to further condense information of cycle life for each battery. A simple variance-based model would, for instance, use $\text{Var}(\Delta Q_{100-10}(V))$ as an input to predict the cycle life for a single battery.

## 3 Model

### 3.1 Physics-Based Model

It is well known that as a lithium-ion battery is cycled, other chemical processes occur in the battery that affects long term cyclability. Notably, the SEI (solid electrode interphase) is formed on the surface of the anode within the first five charging cycles, impeding electron movement in the battery. Although poorly understood, it is believed that SEI formation has an impact on capacity loss (von Kolzenberg et al., 2020). Hence it is important to consider the impact of SEI formation when studying capacity loss. In response, we construct a model of the capacity curves in the Severson et al. (2019) dataset via an Arrhenius Law, which is commonly associated with chemical processes akin to SEI formation (Wang et al., 2011; Spotnitz, 2003; Broussely et al., 2001).

We now introduce an equation describing the process of capacity loss. Wang et al. (2011) have shown that for different charge rates, the true capacity loss $Q_{\text{loss}}$ can be approximated as[1]

$$\hat{Q}_{\text{loss}}(x) = D \exp\left(-\frac{E_a}{RT}\right) x^B \qquad (2)$$

for constants $B$ and $D$, where $\hat{Q}_{\text{loss}}$ represents the percentage of capacity loss, $E_a$ is the activation energy, $R$ the gas constant, $T$ the absolute temperature, and $x$ the cycle number. Note that the temperature dependence in the exponential term resembles the Arrhenius Law, which is commonly associated with a thermally activated chemical process, such as SEI layer formation during cycling (Wang et al., 2011; Spotnitz, 2003; Broussely et al., 2001). Equation (2) aims to account for SEI formation that produces undesired side effects in batteries. In addition, there is also a power law with respect to the cycle number, consistent with previous findings of the rate of lithium consumption at the negative electrode (Wang et al., 2011; Spotnitz, 2003; Wright et al., 2002; Broussely et al., 2001). In short, this equation seeks to describe SEI formation and is consistent with the resulting mechanism of active lithium consumption in the presence of the SEI layer.

We adapt Equation (2) in three ways, while maintaining its physical interpretation. In our dataset, we observe that cells typically show an initial capacity loss even at cycle 0, which can be attributed to various factors, including calendar aging during storage and formation cycles. To account for this, we introduce a constant term $C$ that represents this initial capacity loss. We use the nominal capacity as our reference than the measured capacity at cycle 0, as this provides a consistent baseline across all cells and allows for more robust comparison between different cells, particularly when initial conditions may vary due to storage time or formation cycles.

Next, we observe that the average temperatures reached during testing did not vary greatly across cycles. Treating $T$ as invariant allows us to incorporate the original exponential factor from Equation (2) into the constant $D$. This has the advantage that $E_a$ need not be known anymore.

---

[1]Two adaptations have been made to the original Equation (2) from Wang et al. (2011). Firstly, we replaced the dependency on time with a dependency on cycle number. This is acceptable since all cycles last equal amounts of time. Secondly, we changed the notation of the parameters to not conflict with notations we introduce next.

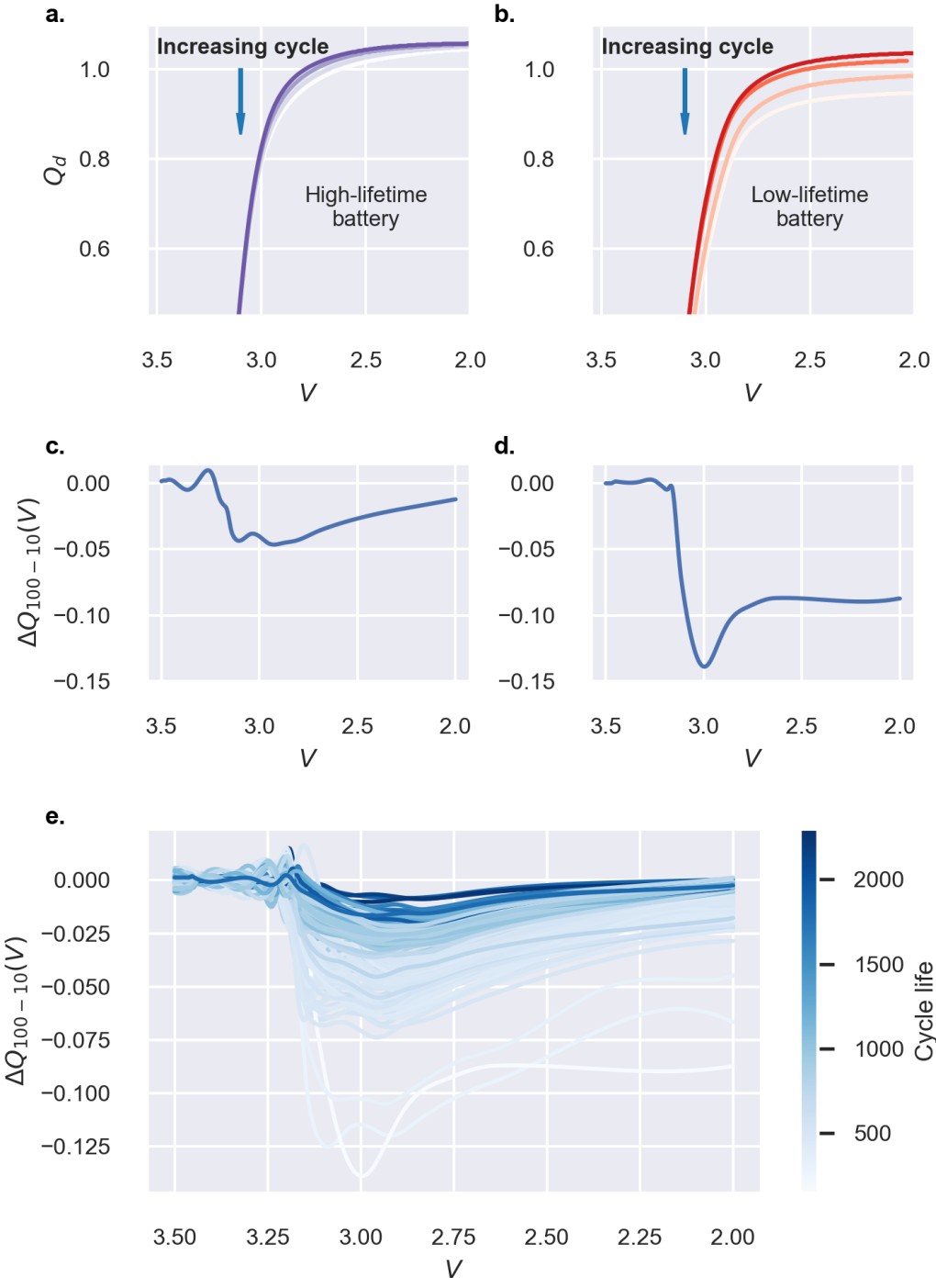

Figure 2: (a, b) Evolution of the discharge-voltage curve over cycles for two batteries with different cycle lives. Curves from cycles evenly spaced between 1 and 100 are plotted and distinguished by saturation. As cycle number increases, the curve progressively sags more for the battery with lower lifetime. (c, d) $\Delta Q_{100-10}(V)$ for the same two batteries. (e) $\Delta Q_{100-10}(V)$ plotted for all batteries in the dataset, with shade corresponding to cycle life.

Lastly, we note that typical values of the constant $D$ have a very large order of magnitude, which introduces numerical instability and overflow issues in further calculations. We mitigate this by predicting $A \equiv \ln D$ instead of $D$ itself.

Thus, we may reduce Equation (2) to one of the form given by Equation (3). By defining $C$ to be equal to the initial capacity loss $Q_{\text{loss}}(0)$, we arrive at a functional form with just two parameters, $A$ and $B$.

$$\hat{Q}_{\text{loss}}(x) = e^A x^B + C \tag{3}$$

We then fit the capacity loss curves in Severson et al. (2019) to Equation (3) via least squares regression. Figure 3(a-c) illustrates three capacity loss curves with substantially different cycle lives and their best fit curves. Not only are the curves themselves a good fit, but the cycle lives as predicted by our best fit curves are remarkably similar to the actual cycle lives. Cycle life $\ell$ is the point where $\hat{Q}_{\text{loss}}(\ell) = 0.2$, calculated from the best fit curve as

$$\ell = [e^{-A}(0.2 - C)]^{1/B}, \tag{4}$$

where 0.2 is used as the threshold capacity loss indicating end of life. Figure 3(d) plots true cycle life against predicted cycle life for all batteries in the dataset and demonstrates high goodness of fit, with $R^2 = 0.994$ and a RMSE of 28.6 cycles. Additional evidence of goodness of fit can be found in Appendix A.

As seen in Figure 3(d) and Appendix A, our equation very accurately models the cycle lives of batteries, and henceforth we assume that the ground truth of capacity loss follows Equation (3), given values of $A$ and $B$. In other words, we assume that $\hat{Q}_{\text{loss}} \approx Q_{\text{loss}}$.

Now the other half of our hybrid model involves training a self-attention model to predict $\hat{A}$ and $\hat{B}$ from cycle input data of the first 100 cycles, which will be explained in detail in the section below. Call this model $f$. Then given a vector $\mathbf{x}$ of early-cycle statistical quantities, our output variables are $(\hat{A}, \hat{B}) = f(\mathbf{x})$ and our predicted cycle life is $\hat{\ell} = [e^{-\hat{A}}(0.2 - C)]^{1/\hat{B}}$. This model is illustrated in Figure 4.

## 3.2 Self-Attention for Regression

We endeavor to predict the parameters of the capacity loss curve based on early-cycle data, when the full capacity loss curve is not known. When limited to early-cycle capacity loss data, we cannot fit and extrapolate the full curve with good fidelity using least squares. Instead, we employ a single self-attention block to learn complex, nonlinear relations between the capacity loss curve and other measurements available during a battery's early operation. As presented in Nguyen et al. (2022), there exists an equivalence between the self-attention mechanism and a support vector regression (SVR) problem formulation. This implies that any problem where an SVR model performs well can potentially be improved by employing an attention-based architecture.

Let the input sequence of $N$ features derived from early-cycle operation data for the $i$-th battery be denoted $\mathbf{z_i} := [z_i^{(1)}, \ldots, z_i^{(N)}]^T \in \mathbb{R}^{N \times 1}$, where each $z(j)_i$ represents a measurement feature such as voltage, current, and temperature . We compute the standard query matrix $\mathbf{Q}$, key matrix $\mathbf{K}$, and value matrix $\mathbf{V}$ from the self-attention mechanism via the following transformations:

$$\mathbf{Q}_i = \mathbf{z}_i \mathbf{W}_Q^T$$
$$\mathbf{K}_i = \mathbf{z}_i \mathbf{W}_K^T$$
$$\mathbf{V}_i = \mathbf{z}_i \mathbf{W}_V^T,$$

where weight matrices $\mathbf{W}_Q$, $\mathbf{W}_K \in \mathbb{R}^{D \times 1}$ and $\mathbf{W}_V \in \mathbb{R}^{D_v \times 1}$ are learnable layers, $D$ is a hyperparameter determining the embedding dimension, and $D_v$ is the output dimension. Note that for the multi-output regression task of predicting two variables, $D_v = 2$. We define the self-attention output $\mathbf{H}_i$ as:

$$\mathbf{H}_i = softmax\left(\frac{\mathbf{Q}_i \mathbf{K}_i^T}{\sqrt{D}}\right)\mathbf{V}_i := \mathbf{A}_i \mathbf{V}_i$$

where the *softmax* function above is applied to each row of the matrix $\frac{\mathbf{Q}_i \mathbf{K}_i^T}{\sqrt{D}}$ to obtain the attention matrix $\mathbf{A}_i$. To collapse the output $\mathbf{H}_i \in \mathbb{R}^{N \times D_v}$ to a vector, we append an averaging layer to the self-attention mechanism that takes the mean along the columns of $\mathbf{H}_i$:

$$\mathbf{y}_i := \mathbf{H}_i^T \mathbf{m} = \begin{bmatrix} \hat{A}_i \\ \hat{B}_i \end{bmatrix}$$

where $\mathbf{m} = [\frac{1}{N} \ldots \frac{1}{N}]^T \in \mathbb{R}^{N \times 1}$. The result $\mathbf{y}_i \in \mathbb{R}^{D_v \times 1}$ is the vector of predicted parameters for the capacity loss curve. These parameters are then used to predict cycle life:

$$\hat{l}_i = [e^{-\hat{A}_i}(0.2 - C_i)]^{1/\hat{B}_i}. \tag{5}$$

### 3.3 Feature Selection

Finally, we select the best features for prediction from the available cycle summary features and discharge curve characteristics described in Section 2. From the cycle summary data, we extract capacity fade characteristics by calculating slopes over different cycle windows. From the discharge-voltage curves $Q_d(V)$, we derive statistical features including the variance, minimum, and mean of the $\Delta Q$ curves, following the approach introduced by Severson et al. (2019). Additionally, we calculate logarithmic transformations of these quantities to better capture the nonlinear relationships in the data.

Our main task is to carefully select the features that are most closely related to our target variables, $\hat{A}$ and $\hat{B}$. To accomplish this, we analyze the correlation between each feature and the target variables. We prioritize features with strong positive or negative correlations, as they are more likely to provide accurate predictions. We use Spearman's correlation coefficient to ascertain which features are most correlated with $\hat{A}$ and $\hat{B}$. In our analysis, we select the top five features with the highest correlation coefficients. The results of our correlation analysis using this approach are visually represented in Figure 5 and the full correlation coefficients can be found in Appendix B.

Note that these features are:

1. DeltaQ_logVars: The log of the variance of the $\Delta Q_{100-10}(V)$ curve.

2. DeltaQ_logMin: The log of the min of the $\Delta Q_{100-10}(V)$ curve.

3. DeltaQ_logMean: The log of the mean of the $\Delta Q_{100-10}(V)$ curve.

4. slope_capacity_fade_2_100: Capacity fade rate from cycle 2 to 100. This is the rate that the capacity fade curve decreases from cycle 2 to cycle 100.

5. slope_capacity_91_100: Capacity fade rate from cycles 91 to 100. This is the rate that the capacity fade curve decreases from cycle 91 to cycle 100.

These five features were used to train the self-attention model as explained in Section 3.2.

### 3.4 Model Training

Our interest lies in minimizing the root mean squared error (RMSE) of cycle life predictions, defined for a set of $n$ batteries as:

$$\text{RMSE}_l = \sqrt{\frac{1}{n} \sum_{i=1}^{n} (l_i - \hat{l}_i)^2}, \tag{6}$$

where $l_i$ is the true cycle life of the $i$-th battery. However, given the complexity of the expression for cycle life, attaining optimal convergence is a nontrivial numerical task. To this end, we employ a two-stage procedure of coarse training on curve parameters followed by fine-tuning on cycle life.

In the first stage, we train on the RMSE of parameter loss, defined as

$$\text{RMSE}_p = \sqrt{\frac{1}{n} \sum_{i=1}^{n} \left( w_A(A_i - \hat{A}_i)^2 + w_B(B_i - \hat{B}_i)^2 \right).} \tag{7}$$

where $w_A$ and $w_B$ are tunable hyperparameters. The parameter loss function is smoother and leads to fewer numerical issues than Equation (6), producing stable results when training from a random initialization. In contrast, we observe that training only with the original cycle life loss function leads to exploding gradients and inconsistent behavior.

However, parameter loss is not always indicative of the accuracy of cycle life predictions. Due to the high nonlinearity of the capacity loss curve, it is possible for two sets of parameter estimates to incur equal parameter loss but produce dramatically different cycle life predictions. Consequently, we follow up coarse training with a fine-tuning stage under low learning rate that trains on cycle life loss.

The two-stage training procedure can be summarized as follows:

1. Coarse training on parameter loss (Equation (7)). This stage guides the model from a random initialization toward approximate parameters, eliminating numerical issues that would otherwise occur in Stage 2.

2. Fine-tuning on cycle life loss (Equation (6)). As many combinations of $\hat{A}$ and $\hat{B}$ can lead to similar parameter losses, this stage tunes the approximate parameters from Stage 1 to produce the most accurate cycle life predictions.

Training took place with 800 epochs and a learning rate of $10^{-3}$ in step 1 and with 3000 epochs and a learning rate of $5 \times 10^{-5}$ in step 2. The choice of optimal hyperparameters are based upon the existing literature, and such parameters can be consulted in the Github repository associated with our work, available upon request. A full comparison of the relevant training curves can be found in Appendix C and Appendix D.

## 4   Results

We initiate the model training phase with a baseline model using a regularized linear framework. For the baseline model, we take sklearn's ElasticNet model that performed the best in Severson et al. (2019). However, to ensure comparable results, we train the baseline model on the features chosen in Section 3.3. We further change the output of the baseline model to predict $\hat{A}$ and $\hat{B}$ in order reconstruct the entire capacity curve. The results of the baseline model are shown in Figure 6.

For hyperparameter tuning, we vary `alpha` and `l1_ratio` on a logscale over $10^0$ and $10^1$ and $10^{-5}$ and $10^2$, respectively. We find that the best model had a primary test RMSE of 398.98 cycles and a secondary test RMSE of 455.4 cycles. We see that in this case, the linear assumptions of elastic net limit its ability to capture nonlinear degradation patterns and parameterize capacity loss curves.

Next, we trained the hybrid model (also referred to as the self-attention model) to predict the parameters $\hat{A}$ and $\hat{B}$. The results of our model evaluation are presented in Figure 6, which compares the performance of the elastic net baseline with that of the self-attention model in terms of the train RMSE, primary test RMSE, and secondary test RMSE. Results from the self-attention model are divided into two stages, Stage 1 and Stage 2, corresponding to the two phases of training.

Note that the errors achieved by self-attention after two stages of training are significantly better than the elastic net baseline; further, they are comparable with the original RMSE errors in Severson et al. (2019). After two training stages, we were able to improve the primary test RMSE from 398.87 to 127.83 cycles and the secondary test RMSE from 455.4 to 179.92 cycles.

Figure 7 illustrates the true versus predicted cycle lives (a) as well as true and predicted capacity curves for sample batteries (b-d) using the fully trained self-attention model. We notice that the predicted capacity

loss curves do indeed fit the behavior of the ground truth curves across train, primary test, and secondary test batteries. Further, recall that the full model in Severson et al. (2019) utilizing multiple features achieves a primary test RMSE of 118 and a secondary test RMSE of 214. Our primary test error is quite similar, but our secondary test error improves upon theirs by 30 cycles (over a 15% improvement). Given that our model captures more information on how a battery degrades over time, we conclude that our model serves as an appealing alternative to predict battery cycle life, especially in cases where one wishes to have the flexibility to consider a different definition of cycle life.

## 5    Conclusion and Future Directions

Our research presents a novel technique of understanding and predicting battery capacity loss curves using a physics-informed model. Our focus on reconstructing these curves rather than on just cycle life prediction offers distinct advantages by providing more robust and interpretable predictions without sacrificing the accuracy achieved in prior work (Severson et al., 2019). Parameterizing the ground truth capacity loss curves by fitting an equation inspired from an Arrhenius Law, we train a self-attention model that recovers these parameters and reconstructs the full capacity curves, achieving similar errors to Severson et al. (2019). This approach is flexible to the definition of end of life, offering the advantage of predicting the cycle at which any percent of capacity is lost.

However, there are a few other directions that potentially further improve upon our work. Notably, the battery cycle data we used is a time series, and hence utilizing entire time series as prediction inputs is one way to incorporate more information and features. This method could potentially improve results given the correct training policy, hyperparameters, and machine learning architectures.

Our method not only provides precise capacity loss forecasts but also incorporates knowledge of the mechanisms underlying battery degradation. The results of this study provide support for the effectiveness of hybrid models, since they combine the best aspects of data-driven and physics-based methodologies. This has the potential to ultimately improve battery life estimation for use in electric vehicles and other socially significant applications.

### Data Availability

The datasets used in this paper are available at https://data.matr.io/1.

### Code Availability

The Github repository can be found at `https://github.com/nathan99sun/HybridPred`.

### Disclosures

C.D Nicolae, S. Sameer, N. Sun, and K. Yan are listed as inventors on a patent application related to this work.

### Acknowledgments

This research was conducted at UCLA as part of the RIPS 2023 REU and is supported by IPAM and the Toyota Research Institute. The authors are grateful to Tan Nguyen and Alex Pham for their mentorship and continued guidance. They also thank Lingyun Ding for fruitful discussions and his continued input. A special thanks to Susana Serna for her continued guidance and feedback on navigating this project. Lastly, the authors are grateful to the anonymous referees for their excellent feedback and comments.

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

## A    Fitted Capacity Loss Curves

This section presents the fitted capacity loss curves against the true capacity loss $Q_{\text{loss}}$. One may be concerned that the fitted capacity loss curves only capture the cycle life instead of the full process of capacity loss. Our reconstruction of the capacity loss curves demonstrate goodness of fit to the entire capacity loss curve, with an average $R^2$ of 0.976 (standard deviation 0.0644) across all batteries in the dataset. Figure 8 plots the $R^2$ of all fitted capacity loss curves compared to the true curves for all batteries, and Figure 9 shows the fitted vs true capacity loss curves for all batteries.

## B    Feature correlation scores with parameters $\hat{A}$ and $\hat{B}$

This section presents the correlation scores between the candidate features and equation parameters $\hat{A}$ and $\hat{B}$. The scores of the five features with the highest correlation are bolded in Table 1. A detailed discussion of feature extraction methodology and their physical significance is provided in Section 2.

| Feature Name | Feature Description | $\hat{A}$ | $\hat{B}$ |
|---|---|---|---|
| DeltaQ_logVar | Variance of logarithmic changes in capacity difference | **0.46** | -0.23 |
| DeltaQ_logMin | Minimum logarithmic change in capacity difference | **0.48** | -0.25 |
| DeltaQ_logMean | Mean logarithmic change in capacity difference | **0.48** | -0.25 |
| DeltaQ_logSkew | Skewness of logarithmic changes in capacity difference | 0.28 | -0.18 |
| DeltaQ_logKurt | Kurtosis of logarithmic changes in capacity difference | 0.34 | -0.20 |
| Slope_capacity_fade_2-100 | Slope of capacity fade between cycles 2 and 100 | **-0.50** | **0.56** |
| Slope_capacity_fade_91-100 | Slope of capacity fade between cycles 91 and 100 | **-0.57** | **0.53** |
| QD_Max_2 | Maximum discharged capacity relative to initial capacity | 0.32 | -0.15 |
| QD_2 | Discharged capacity measured at the second cycle | 0.22 | -0.18 |
| Intercept_capacity_fade_2-100 | Intercept of the capacity fade between cycles 91 and 100 | -0.30 | 0.26 |
| Intercept_capacity_fade_91-100 | Intercept of the capacity fade between cycles 91 and 100 | 0.15 | -0.12 |
| Min_IR | Minimum internal resistance measured during cycling | -0.20 | -0.21 |
| IR_100-2 | Difference in internal resistance between cycles 100 and 2 | 0.19 | -0.17 |

Table 1: Feature correlation scores with $\hat{A}$ and $\hat{B}$

## C Training curves in two-stage training procedure

Figure 10 presents the training curves after each stage of the two-stage training procedure described in Section 3.4. As expected, the curves after Stage 1 provide a rough estimate of the capacity loss curve, and is refined after Stage 2 of training.

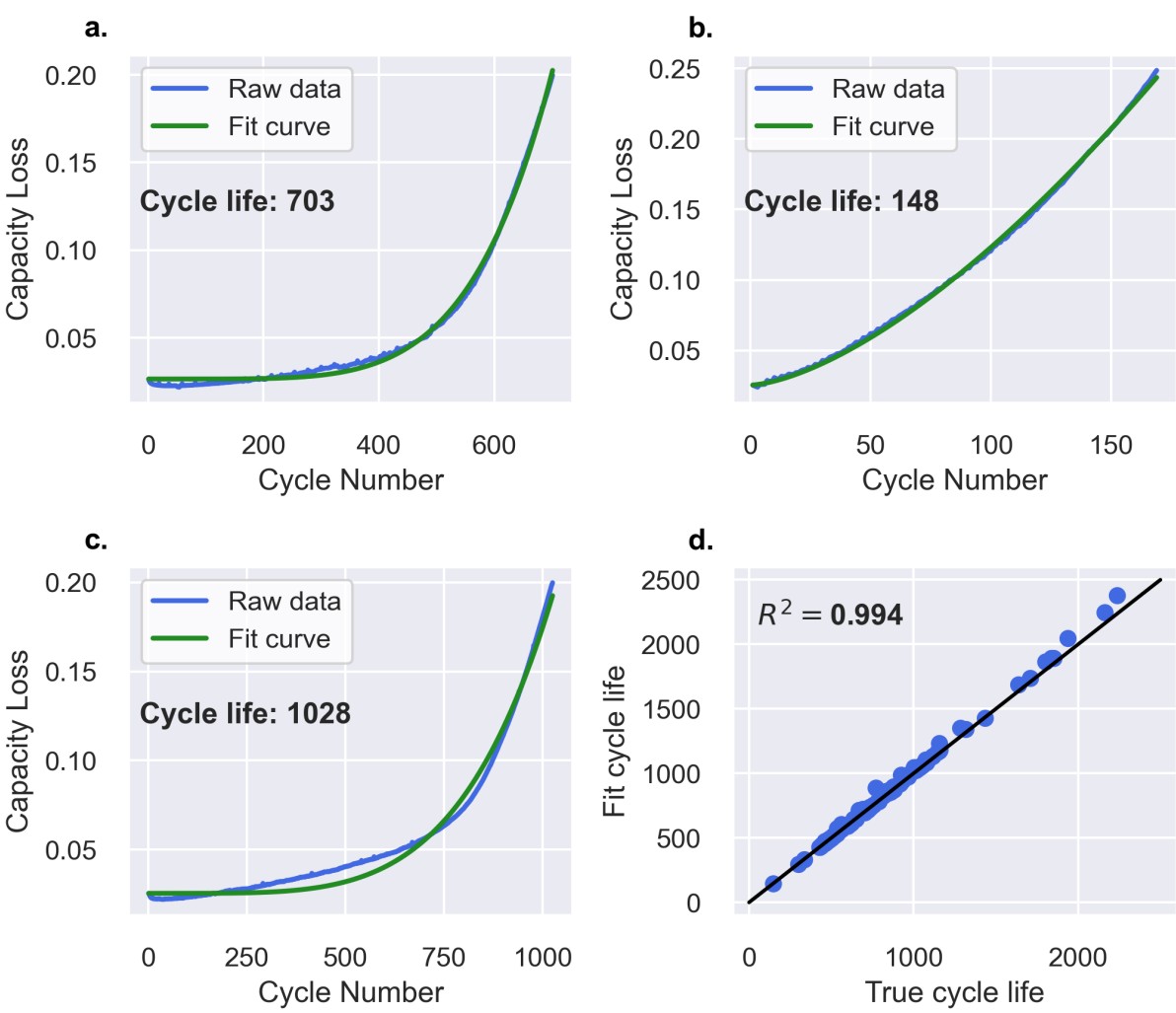

Figure 3: (a-c) Capacity loss model fit to three capacity loss curves using least-squares. The three curves reflect batteries with substantially different lifetimes, demonstrating the ability of the model to generalize. $R^2$ for each individual battery is displayed, and average $R^2$ across all 124 batteries in the dataset is 0.976. (d) Cycle lives derived from the fitted capacity loss curves, plotted against true cycle lives. We observe $R^2 = 0.994$, demonstrating high goodness of fit.

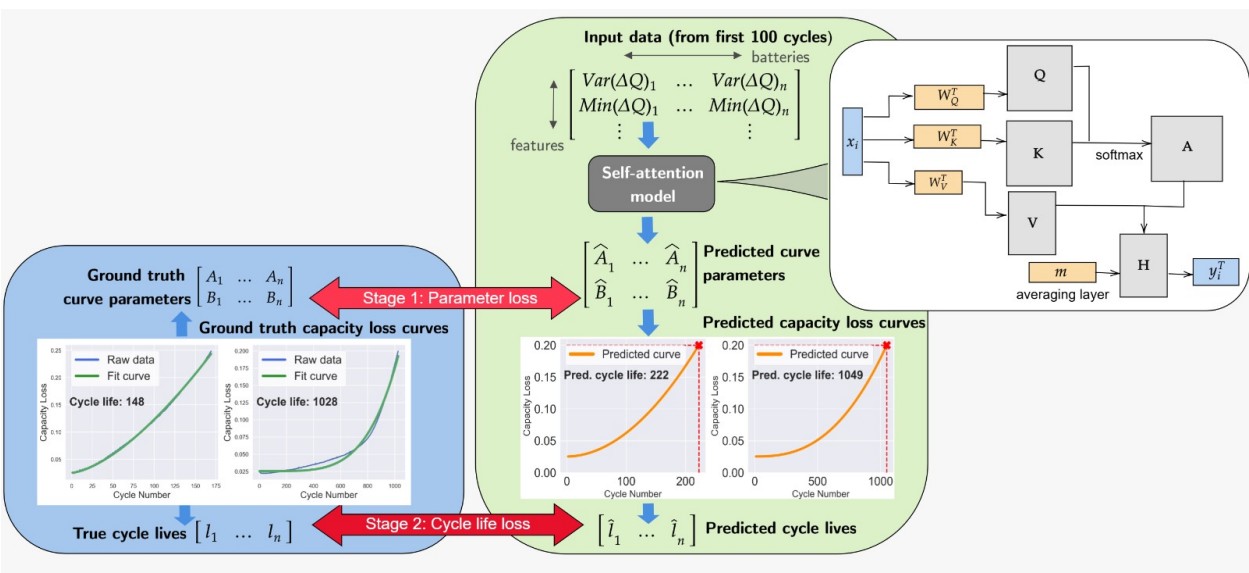

Figure 4: Schematic of the physics-based model. One half utilizes an Arrhenius Law-inspired model to capture capacity loss curves. The other half utilizes a self-attention layer to predict Arrhenius Law parameters from early-cycle data.

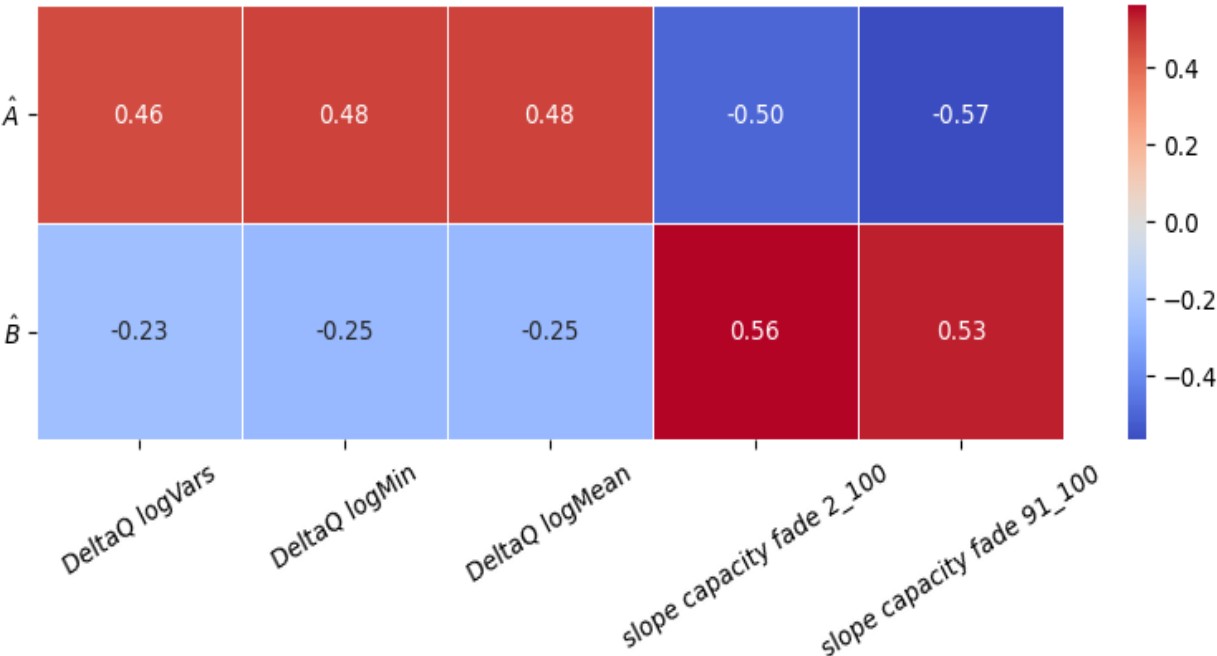

Figure 5: Feature correlation scores of the five features most correlated with $\hat{A}$ and $\hat{B}$.

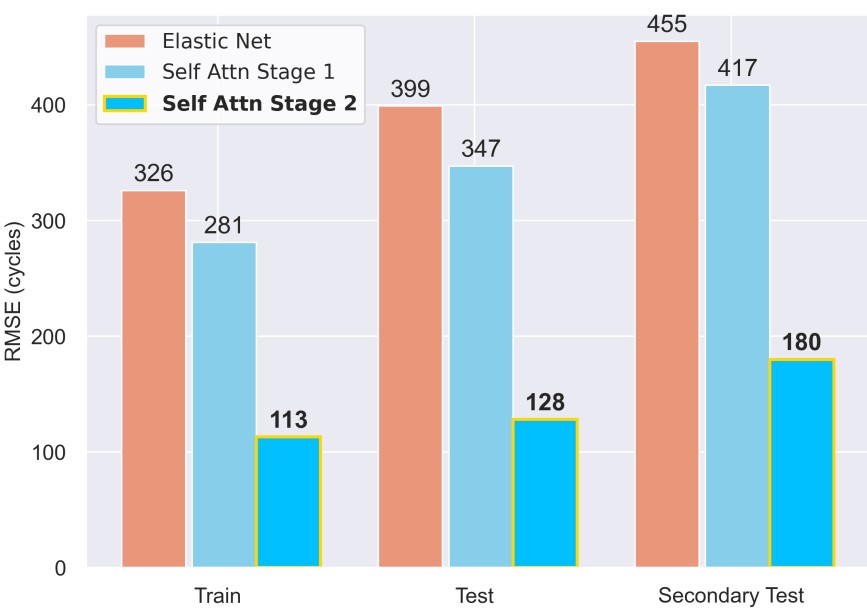

Figure 6: Comparison of RMSE for cycle life predictions made by the elastic net baseline model, the self-attention model after the first stage of training, and the self-attention model after both stages of training.

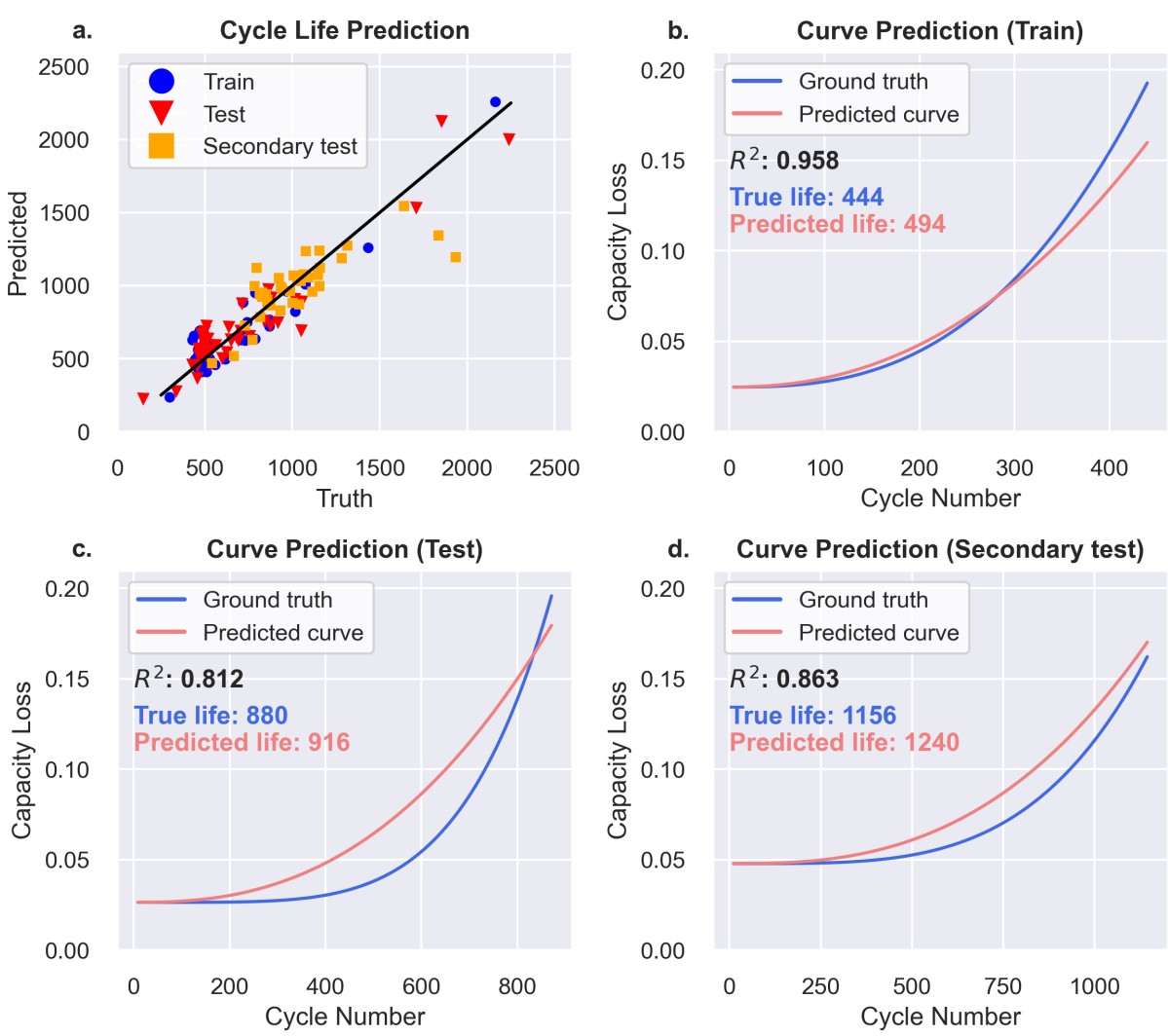

Figure 7: (a) Predicted versus true cycle lives using the self-attention model. (b-d) Predicted versus true capacity curves using the self-attention model for example batteries from the (b) train, (c) test, and (d) secondary test datasets.

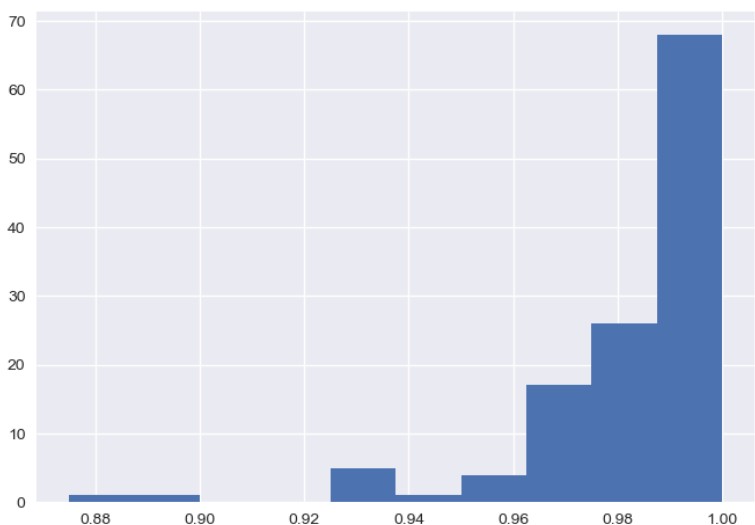

Figure 8: Distribution of $R^2$ for each fitted capacity loss curve, excluding one outlier at 0.295.

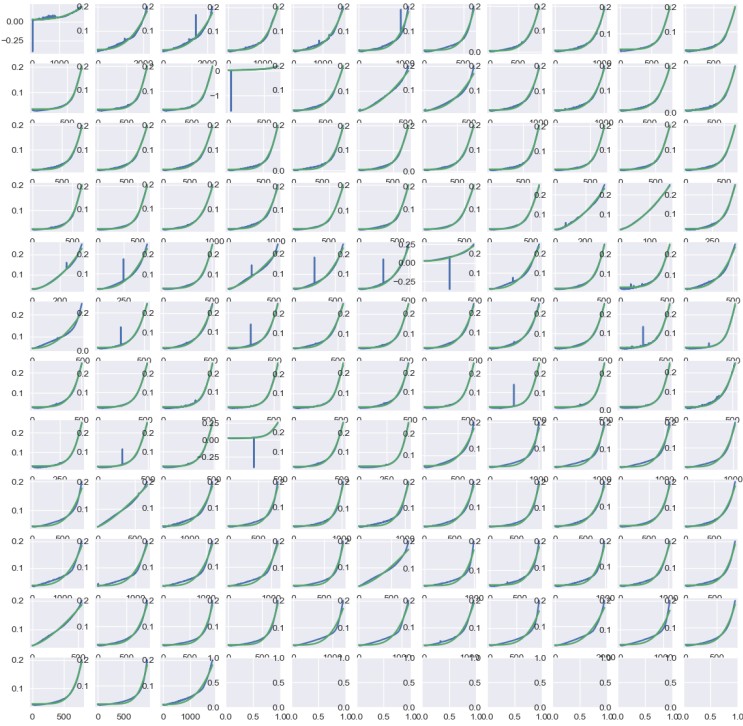

Figure 9: Fitted vs true capacity loss curves via Equation 3.

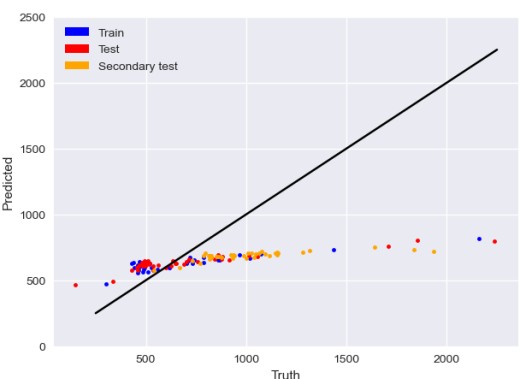

(a) Predicted and true cycle lives after Stage 1

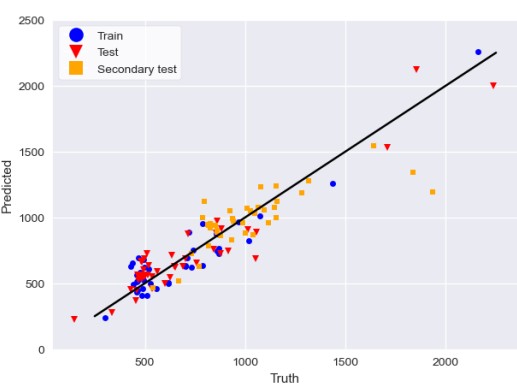

(b) Predicted and true cycle lives after Stage 2

Figure 10: Predicted and true cycle lives after each stage of the two-stage training procedure.

## D Training curves in two-stage training procedure

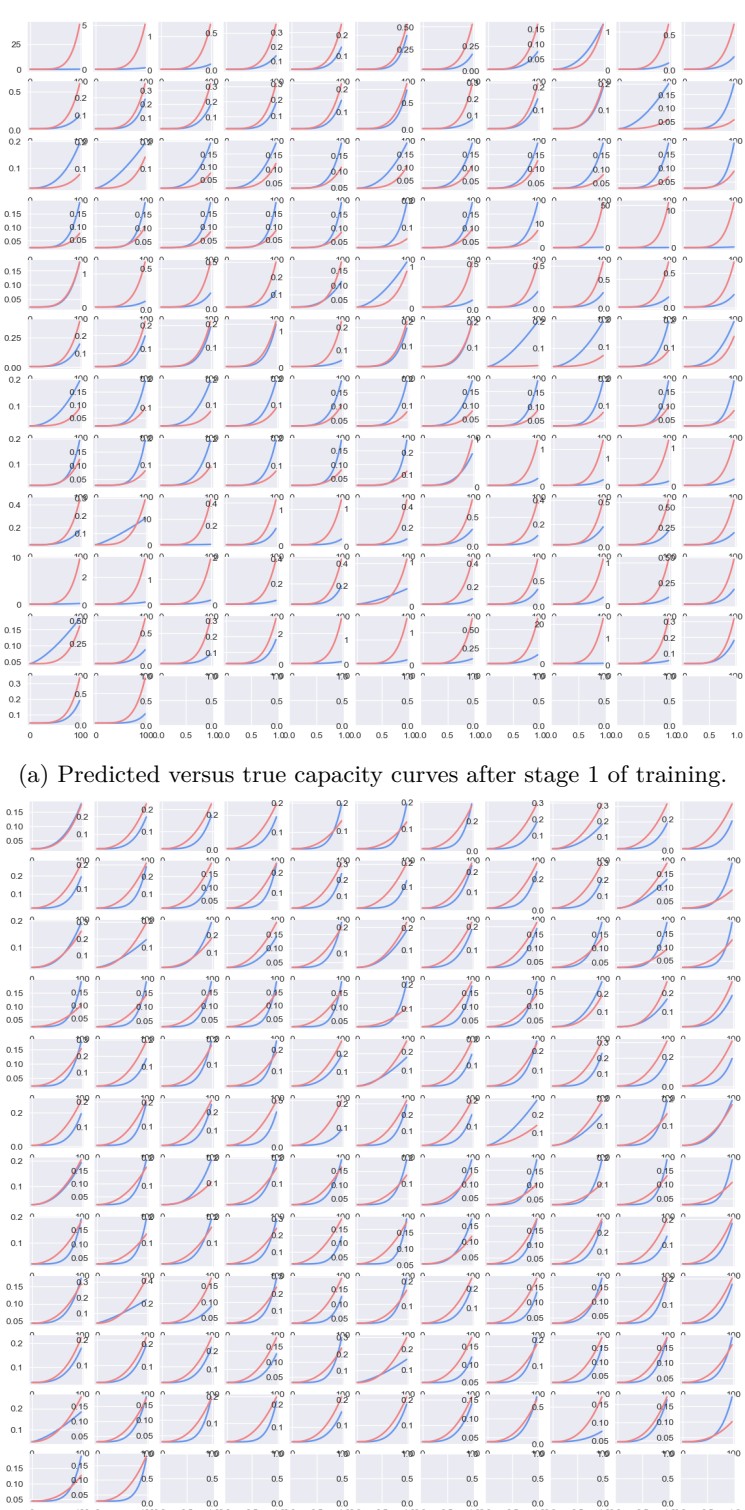

(a) Predicted versus true capacity curves after stage 1 of training.

(b) Predicted versus true capacity curves after stage 2 of training.

Figure 11: Predicted versus true capacity curves after each stage of the two-stage training procedure.

