# OpenReview forum: "Optimizing Cycle Life Prediction of Lithium-ion Batteries via a Physics-Informed Model"
_TMLR — Accepted by TMLR_

### Review · Reviewer_FBYN · 2024-10-24

**Summary Of Contributions:**

Optimizing cycle life prediction of lithium-ion batteries via a physics-informed model presents a modeling approach to predicting capacity fade as a function of cycle time. The authors first motivate the use of a structured (parameterized) model using all data showing that good fits can be obtained then transition to predict the parameters of the model using other derived features measured only during early cycling. They chose an approach based on self-attention for the later. The approaches are evaluated on a large LiFePO4 graphite dataset.

**Audience:**

Yes

**Claims And Evidence:**

Yes

**Requested Changes:**

Introduction:
- The phrase "difficult to fine-tune" should be clarified as I was not led to believe that the authors meant this in a numerical sense but I also didn't follow the point they are trying to make.

Physics-based Model:
- The phrase "Note that the temperature highly resembles..." didn't make sense to me but I wonder if there is a typo? Please clarify.
- I found "accounts for SEI formation" to be too strong of a statement and suggest rephrasing to something along the lines of "aims to account for SEI formation"
- I think the authors mean to say that they predict e^B not lnB such that A is the contributions of the temperature term and e^B? It is confusing that the exponent in equation 2 for cycle life in eqn 2 is z and then B in eqn 3 as it appears that B != B.
- "Ground truth capacity loss" is confusing. I think the authors are saying that \hat{Q}(0)_loss != 0 because of calendar aging? This is not a "reduction" of equation 2 but instead of different model specification. It would be good for the authors to comment on the choice of always using the nominal capacity as opposed to the measured capacity at cycle 0 in defining capacity loss.
- In Figure 3, I would argue that the model didn't "generalize" as a different model is fit for each curve using all of the data. However, the parameterization is able to generalize. I think it would be useful for the authors to present RMSE as a function of cycle life to help the reader understand possible biases in the parameterized model.

Self-attention for regression
- I think the ‘x’ in the description of the self-attention section is not the same as ‘x’ in the physics-based model section and therefore should use another term. I also think the authors use a multidimensional sequence of inputs to predict A and B. Please clarify.
- The equation for cycle life sometimes uses Q(0) and sometimes uses C

Feature selection
- The candidate features are not described nor is the procedure for proposing candidates. This discussion should be added.

Model training
- Did you try training using log l_i for improved numerics?

Results
- The results section is missing an analysis of using a linear model to predict A and B and comparing. As presented, I feel the contribution of the physics-informed model vs. the attention-based parameterization is unclear. I think understanding the relative contributions of the framework is critical to the manuscript.
- I find the results in figure 6 confusing - the linear model is worse than Severson et al. because it only considers the five features selected in the Spearman’s analysis?

Small comments:
- In the data processing section, there is a typo where "three forms" of data are mentioned before a list of four items

**Strengths And Weaknesses:**

Strengths:
- The combination of physics-informed and data-driven cycle life prediction feels well motivated.

Weaknesses:
- The exposition of the approach and subsequent evaluation feels incomplete. I provide specific comments in the request changes section.
- I'm somewhat uncertain about TMLR being the correct audience for this work as it feels more relevant to the battery community but I will leave it to the editors to weigh in on this point.

---

### Review · Reviewer_pbnc · 2024-12-03

**Summary Of Contributions:**

This paper sets up a framework to predict the cycle lifetime of lithium-ion batteries. First, a physics-based equation is used to fit the capacity loss as a function of number of life cycles. Then, a self-attention model is trained in a two-stage process to predict the parameters derived from the fit, taking as input features obtained from the early life of the battery (up to cycle 100). The authors show that on a given dataset of lithium-ion battery cells lifetime characteristics, this hybrid physics-attention approach can match or improve predictive performance of competing approaches not based on self-attention.

**Audience:**

Yes

**Broader Impact Concerns:**

not addressed in the manuscript. No further concern on my end.

**Claims And Evidence:**

Yes

**Requested Changes:**

see previous section

**Strengths And Weaknesses:**

**strengths**

- the paper is well written and easy to follow, the problem statement is well presented, and the description of the components of the hybrid approach (physics + self-attention) is also clear
- the topic of using machine learning-augmented approaches towards prediction of physical properties (of batteries, in this case, but potentially of other physical systems as well) may be of interest to the audience of this journal

**weaknesses**

While the problem and approach are well introduced, the paper loses sharpness when presenting architectural details and results.

- Section 3.2: the architecture is not entirely clear. In my understanding, the authors implement self-attention mechanism without several of the common building blocks found in current transformers (e.g., normalization, shortcuts). Does this model consists of a *single* self-attention block, that is, it consists of a total of 3 linear modules (Q, K, and V) plus an averaging operation (m)? Are other components in use? It would be useful to have a clear schematic of the architecture, maybe added to Figure 4.
- Section 3.3: five features are selected to train the self-attention model, based on their correlation score with the two physics-based parameters. The authors should report which features were originally taken into considered and what was their correlation. Also, the select five features are not properly defined, only named, so their meaning is unclear.
- Section 3.4: the training process is not well described. How many examples and training steps were taken? Can the authors provide the training / validation curves, to verify that the model is training correctly?
- Section 4: the authors present values for a baseline "similar to the model that performed the best in Severson et al." [1] which is based on ElasticNet. It is not clear how these new baseline values compare with [1], as the authors use RMSE but [1] reports test error in the main text. Similarly, it is later stated that a "the full model in Severson et al. (2019) utilizing multiple features achieves a primary test RMSE of 118 and a secondary test RMSE of 214." Can the authors set up a direct comparison of the baselines and clarify what models taken from [1] they compare against?
- Related to the previous comment, it is not clear why Figure 6 reports data for that particular baseline but not for the strongest one provided by [1]. Can the authors clarify this point?
- Overall, my understanding is that the predictive capabilities on cycle lifetime of the proposed hybrid approach appear to be on par or, in some cases, moderately better than the best prediction obtained with the baseline approach. The authors stress that their hybrid approach provides not only a single estimate of lifetime, but the expected capacity loss curve as well. I notice from Figure 7 that there are definitely discrepancies between the estimated model and the ground truth. Can the authors outline some specific scenarios where knowledge of the capacity loss curve would be beneficial? In such scenarios, what discrepancy between prediction and actual behavior would be acceptable?

** other comments **

- typo in Section 2: "For each cell, three forms of data are recorded" is followed by four types of data
- the authors mention that a Github repository is available upon request. I recommend adding the link to the paper, after the anonymous review phase has been completed.


[1] Severson, et al. Data-driven prediction of battery cycle life before capacity degradation. Nature Energy, 4(5):383–391, 2019.

---

### Review · Reviewer_6Pu7 · 2025-02-07

**Summary Of Contributions:**

The paper proposes and evaluates a hybrid approach to predict the battery cycle life time and the battery capacity loss curves for commercial lithium-ion batteries. The hybrid approach uses a combination of a physics-based equation and a self-attention model. The hybrid approach seems novel, although batteries as an application context is out of my competence domains.

The paper contains two main novelties, as far as I can see:
* The combination of physics-based equation and an attention mechanism.
* The prediction of not only the battery cycle lifetime, but also the capacity loss curve over time (charging cycles).

**Audience:**

Yes

**Claims And Evidence:**

Yes

**Requested Changes:**

I think the authors have done a good job in addressing the other reviewers' comments. Thus, I have no required changes.

**Strengths And Weaknesses:**

**Strengths**
+ In general, a well written and easy to read paper.
+ Good selection of evaluation metrics.
+ In general, a good evaluation of the proposed model.
+ The results show good prediction accuracy after the second-stage training.
+ Good appendix with complimentary information.
+ Good and up-to-date list of references and related work.

**Weaknesses**
- I'm surprised about the correlations between A_hat and B_hat, and features are relatively low, but the model still seems to work well. Maybe a deeper analysis and discussion of that?
- The evaluation is done using only one dataset, with relatively few batteries, all of them low voltage and relatively low capacity. I know that the availability of datasets always is an issue, but it would have been nice to at least have a discussion about the applicability of the results for high-voltage and high-capacity batteries.

---

### Author Response · Authors · 2025-01-28
**Responses to Reviewer pbnc**

Section 3.2: the architecture is not entirely clear. In my understanding, the authors implement self-attention mechanism without several of the common building blocks found in current transformers (e.g., normalization, shortcuts). Does this model consists of a single self-attention block, that is, it consists of a total of 3 linear modules (Q, K, and V) plus an averaging operation (m)? Are other components in use? It would be useful to have a clear schematic of the architecture, maybe added to Figure 4.
- The model consists of a single self-attention block with an appended averaging operator; no other components in use. Figure 4 has been updated.

Section 3.3: five features are selected to train the self-attention model, based on their correlation score with the two physics-based parameters. The authors should report which features were originally taken into consideration and what was their correlation. Also, the select five features are not properly defined, only named, so their meaning is unclear.
- Added the full correlation scores to the appendix. The definitions and scores of selected features are also added. We’ve further provided more meaning to the selected five features in Section 3.3.

Section 3.4: the training process is not well described. How many examples and training steps were taken? Can the authors provide the training / validation curves, to verify that the model is training correctly?
- Included in the appendix.

Section 4: the authors present values for a baseline "similar to the model that performed the best in Severson et al." [1] which is based on ElasticNet. It is not clear how these new baseline values compare with [1], as the authors use RMSE but [1] reports test error in the main text. Similarly, it is later stated that a "the full model in Severson et al. (2019) utilizing multiple features achieves a primary test RMSE of 118 and a secondary test RMSE of 214." Can the authors set up a direct comparison of the baselines and clarify what models taken from [1] they compare against?
Related to the previous comment, it is not clear why Figure 6 reports data for that particular baseline but not for the strongest one provided by [1]. Can the authors clarify this point?
- In response to the above two points: we take the ElasticNet model from Severson et al. and train it to predict A and B; this serves as our baseline. To ensure a direct comparison, we wanted to ensure that both our model and baseline accounted for the physics-based insights in the equation presented in Section 3.1, and hence our baseline adapts the ElasticNet model from Severson to include the features we chose and accounts for the physics-based model for a direct comparison to our results. We’ve made this clearer in the paper, and can also include Severson’s actual RMSEs if desired.

Overall, my understanding is that the predictive capabilities on cycle lifetime of the proposed hybrid approach appear to be on par or, in some cases, moderately better than the best prediction obtained with the baseline approach. The authors stress that their hybrid approach provides not only a single estimate of lifetime, but the expected capacity loss curve as well. I notice from Figure 7 that there are definitely discrepancies between the estimated model and the ground truth. Can the authors outline some specific scenarios where knowledge of the capacity loss curve would be beneficial? In such scenarios, what discrepancy between prediction and actual behavior would be acceptable?
- The most immediate: currently the cycle life of a battery is defined as the point where the capacity drops to 80% of its maximum capacity. However, the exact number can vary due to battery manufacturer and quality, in addition to the conditions the battery is in (for example, 85% is also used by battery manufacturers; and if one wishes to account for a battery being in non-ideal conditions, one may want to consider a higher percentage of maximum capacity). If one wishes to consider an alternative threshold to calculate cycle life, knowledge of the capacity curve can be immediately used to predict cycle life, while other models need to be retrained. In short, knowledge of the entire capacity curve gives us cycle lives for all capacity drops, giving more information on the evolution of the lifetime of a battery. We imagine that a discrepancy in the magnitude as the RMSEs reported in Figure 6 would be acceptable.

** other comments **

typo in Section 2: "For each cell, three forms of data are recorded" is followed by four types of data
- Fixed.

the authors mention that a Github repository is available upon request. I recommend adding the link to the paper, after the anonymous review phase has been completed.
- Yes, we will include the link after the review phase.

Many thanks to Reviewer pbnc for many helpful suggestions and feedback.

---

### Author Response · Authors · 2025-01-28
**Responses to Reviewer FBYN**

The phrase "difficult to fine-tune" should be clarified as I was not led to believe that the authors meant this in a numerical sense but I also didn't follow the point they are trying to make.
- Fixed, the phrase is further elaborated in the paper. It is difficult to tune those models, as their performance depends heavily on hyperparameter optimization, feature selection, and the quality of the training data.

The phrase "Note that the temperature highly resembles..." didn't make sense to me but I wonder if there is a typo? Please clarify.
The sentence is revised to “Note that the temperature dependence in the exponential term resembles the Arrhenius Law”.
I found "accounts for SEI formation" to be too strong of a statement and suggest rephrasing to something along the lines of "aims to account for SEI formation"
- Fixed.

I think the authors mean to say that they predict e^B not lnB such that A is the contribution of the temperature term and e^B? It is confusing that the exponent in equation 2 for cycle life in eqn 2 is z and then B in eqn 3 as it appears that B != B.
- Fixed. The notations have been adapted to avoid the conflict.

"Ground truth capacity loss" is confusing. I think the authors are saying that \hat{Q}(0)_loss != 0 because of calendar aging? This is not a "reduction" of equation 2 but instead of different model specification. It would be good for the authors to comment on the choice of always using the nominal capacity as opposed to the measured capacity at cycle 0 in defining capacity loss.
- Yes, the initial capacity loss observed at cycle 0 can be attributed to various factors, including calendar aging during storage and the effects of formation cycle.
- We use the nominal capacity as our reference point rather than the measured capacity at cycle 0, as this provides a consistent baseline across all cells and allows for more robust comparison between different cells. This point has been added in the manuscript.
- Adding the constant is indeed not a model simplification. The wording has been corrected.

In Figure 3, I would argue that the model didn't "generalize" as a different model is fit for each curve using all of the data. However, the parameterization is able to generalize. I think it would be useful for the authors to present RMSE as a function of cycle life to help the reader understand possible biases in the parameterized model.
- Fixed in Section 3.4.

I think the ‘x’ in the description of the self-attention section is not the same as ‘x’ in the physics-based model section and therefore should use another term. I also think the authors use a multidimensional sequence of inputs to predict A and B. Please clarify.
- A new variable (Z) has been introduced to distinguish the two, and the input sequence has been clarified in the paper.

The equation for cycle life sometimes uses Q(0) and sometimes uses C
- Changed to C for consistency.

The candidate features are not described nor is the procedure for proposing candidates. This discussion should be added.
- Candidate features and their correlation scores are added in the appendix.

Did you try training using log l_i for improved numerics?
- Yes, this led to worse results.

The results section is missing an analysis of using a linear model to predict A and B and comparing. As presented, I feel the contribution of the physics-informed model vs. the attention-based parameterization is unclear. I think understanding the relative contributions of the framework is critical to the manuscript.
I find the results in figure 6 confusing - the linear model is worse than Severson et al. because it only considers the five features selected in the Spearman’s analysis? Small comments:
- In response to the two points above: our ElasticNet baseline serves as the linear model that predicts \hat{A} and \hat[B} and comparing. This wasn’t made sufficiently clear in our earlier draft, and we have addressed this in our revised draft.
The linear model is worse than Severson et al. because it is also expected to reconstruct the entire capacity loss curves (while in Severson et al. it only needed to predict the cycle life). To ensure a direct comparison, we borrowed the ElasticNet model used in Severson et al. and trained it to predict \hat{A} and \hat{B}.

In the data processing section, there is a typo where "three forms" of data are mentioned before a list of four items.
- Fixed.

Thank you to Reviewer FBYN for many helpful suggestions and feedback.

---

### Decision · Action_Editor_Jciq · 2025-03-31

**Recommendation:** Accept with minor revision

**Comment:**

All reviewers voted for acceptance. The authors addressed the concerns raised by the reviewers, including clarifying the architecture, feature selection, training, etc. One reviewer (FBYN) asked for one additional figure showing the RMSE as function of the cycle number (as oppose to cycle life) as well as providing evidence that better supports the claim that the capacity fade curve is recovered and provide a comparison to the individual fits vs those recovered using the proposed features. I'd request the authors to provide this additional details in the final version of the paper.

**Audience:**

This is an application paper. It is of relatively narrow interest, but the proposed approach is sound and worth publishing.

**Claims And Evidence:**

All reviewers agree that the claims are supported. This work proposes a hybrid approach to predict the battery cycle life time and the battery capacity loss curves for commercial lithium-ion batteries. The approach combines a physics-based equation and a self-attention model. One reviewer praised the additional details presented in the Appendices. Several reviewers indicated that the presentation was significantly improved.